# COVID-19 Vaccination Trends and Side Effects among Egyptian Hemodialysis Patients: A Multicenter Survey Study

**DOI:** 10.3390/vaccines10101771

**Published:** 2022-10-21

**Authors:** Mohammed Kamal Nassar, Karem Mohamed Salem, Mohamed Elgamal, Sara M. Abdel-Gawad, Samar Tharwat

**Affiliations:** 1Mansoura Nephrology & Dialysis Unit (MNDU), Department of Internal Medicine, Faculty of Medicine, Mansoura University, Mansoura 35511, Egypt; 2Nephrology & Dialysis Unit, Department of Internal Medicine, Faculty of Medicine, Fayoum University, Fayoum 63511, Egypt; 3Chest Department, Faculty of Medicine, Mansoura University, Mansoura 35511, Egypt; 4Mansoura Nephrology and Dialysis Unit (MNDU), Mansoura University, Mansoura 35511, Egypt; 5Rheumatology & Immunology Unit, Internal Medicine Department, Faculty of Medicine, Mansoura University, Mansoura 35511, Egypt

**Keywords:** side effects, Sinopharm, AstraZeneca, vaccine, COVID-19, hemodialysis, Egypt

## Abstract

(1) Background: Vaccination may be a key intervention to prevent infection in chronic hemodialysis (CHD) patients. This study aimed to determine the COVID-19 vaccination status in Egyptian CHD patients and to analyze the safety and detailed side effect profile of the COVID-19 vaccine among these patients. (2) Methods: This survey-based study was conducted on 670 end-stage renal disease (ESRD) patients on CHD from 3 December 2021 to 5 February 2022. Subjects were asked about sociodemographic characteristics, clinical and therapeutic data, in addition to their COVID-19 vaccination status. If the subject had been vaccinated, we inquired about the type of vaccine and the side effects that occurred within a few days after administration of the first and second dose of the COVID-19 vaccine. Additionally, subjects were asked about the onset of side effects (days from vaccination), timing of maximum symptoms, intensity of symptoms and their effect on activity and need for medical attention. (3) Results: The study included 670 CHD patients with a mean age of 50.79 years; 58.1% were females. The vast majority (614; 91.6%) of the studied patients received two doses of the vaccine. Side effects were more commonly reported after the first dose than the second dose. The main side effects reported were generalized weakness/fatigue (56%), headache (43.8%) and fever (40.4%), and sore arm/pain was also reported (29.3%). Adverse events mostly occurred within one day after vaccination and the maximum symptoms usually happened on the second day. The median duration of symptoms was 3 days with a maximum duration up to 5 days. The univariate logistic regression analysis showed that male gender (OR 1.848; (95% CI, 1.242–2.749), *p* = 0.002), age (OR 0.981; (95% CI, 0.969–0.993), *p* = 0.003), smoking (OR 6.067; (95% CI, 3.514–10.475), *p* < 0.001), duration since starting HD (OR 0.998; (95% CI, 0.998–0.999), *p* < 0.001), associated comorbidities (OR 2.202; (95% CI, 1.478–3.281), *p* < 0.001) and prior COVID-19 infection (OR 3.318; (95% CI, 1.952–5.642), *p* < 0.001) were the main determinants of adverse events related to COVID-19 vaccination. (4) Conclusions: our preliminary findings support the favorable short-term safety profile of the COVID-19 vaccine among CHD patients, and hence can reassure both clinicians and patients, as well as further promote COVID-19 vaccine administration among these patients.

## 1. Introduction

The severe acute respiratory syndrome coronavirus 2 (SARS-CoV-2), also known as coronavirus disease (COVID-19), is a major pandemic that is imposing a great burden on healthcare systems all around the world [1]. Patients on dialysis for end-stage kidney disease (ESKD) face unique circumstances that increase their risk of SARS-CoV-2 infection. Most of these patients receive in-center hemodialysis (HD), which requires them to visit the dialysis center three times per week for at least three to four hours, while being surrounded by other patients and staff. As a result, these patients cannot self-isolate [2].

HD patients who develop COVID-19 are at least 30% to 130 % more likely to die than non-chronic kidney disease (CKD) hospitalized individuals [3]. Once infected, HD patients have a greater risk of death due to impaired immune function and a higher prevalence of other well-known risk factors for COVID-19-associated mortality, such as hypertension, obesity, diabetes, and heart disease. To date, there is substantial evidence that non-elderly HD patients have abnormally high death rates, which, in most cases, match or even exceed rates reported in older patients in their respective locations [4,5].

During the subsequent waves of the pandemic, infection prevention measures were gradually implemented in dialysis centers to minimize the rate of transmission [3,6]. Additionally, COVID-19 vaccines appeared to be successful in reducing severe illness and hospitalization [7]. As a result, the COVID-19 vaccine is recommended for HD patients. However, due to a paucity of evidence on the safety and efficacy of SARS-CoV-2 vaccines in HD patients, the medical community has developed some degree of nihilism towards the vaccine [2].

The Egyptian government has devised and executed plans to contain and mitigate this pandemic’s effects on human health and the economy [8]. It initiated a number of programs to increase public awareness of the worldwide epidemiological crisis [9]. Parallel to these initiatives, all citizens are being offered free vaccinations, with priority given to healthcare workers (HCWs) and elderly individuals, particularly those with chronic conditions [10]. In this regard, the government has also made arrangements to obtain vaccinations from various sources to meet its immunization requirements [8].

This study aimed to determine the vaccination status in Egyptian CHD patients and analyze the safety and detailed side effect profile of the COVID-19 vaccine among these patients.

## 2. Materials and Methods

### 2.1. Design

This survey-based study was conducted from 3 December 2021 to 5 February 2022 to evaluate the vaccination status and prevalence of side effects of the COVID-19 vaccine among randomly selected Egyptian CHD patients.

### 2.2. Participants

The study included ESRD patients on CHD for more than 6 months and aged >18 years, who either were vaccinated with the COVID-19 vaccine or not. Vaccinated participants must have received the most recent dosage of the vaccine, either the first or second dose, no more than thirty days before completing the questionnaire. The participants were recruited from different Egyptian HD units in the Dakahlia Mansoura Nephrology and Dialysis Unit (MNDU), Fayom (Fayoum General Hospital, Tamia Hospital, Fayoum Insurance Hospital) and Cairo (Rawas Hemodialysis Center). Patients aged less than 18 years or who had psychiatric or neurological disabilities were excluded from the start.

Patients were approached (in person) and provided complete questionnaire data, as well as examples of commercial names for generic pharmaceutical drugs. The questionnaires were given to patients at respective HD units by the researchers. The questionnaire was self-administered for people who could read and write, and it was delivered by a researcher for those who could not.

### 2.3. Calculation of Sample Size

The online sample size calculator RaoSoft^®^ was used to determine the appropriate sample size. Based on an estimated prevalence of patients on dialysis (264 pmp [11]), the 50% predicted response, 5% margin of error and 95% confidence level, the minimum sample size was 397 participants.

### 2.4. The Questionnaire

The study used a questionnaire with multiple-choice items. The questionnaire was prepared and then refined and revised. It was then provided to the patients in Arabic after being translated. Five nephrology staff members examined the translation, questionnaire design, substance, words, comprehension, and ease of completion in a pilot study to validate the questionnaire. The questionnaire was finalized through an iterative conversation. The questions were written to be as straightforward as possible. Answers were limited to yes/no questions (closed-ended questions). Later, the questionnaire’s reliability was assessed by a group of twenty recently vaccinated HD patients, who completed the questionnaire twice with a minimum interval of two weeks. The provisional instrument’s re-test resulted in significant reliability, with a mean Cohen’s kappa coefficient of 0.84 ± 0.16. The following areas were covered in the survey: sociodemographic characteristics, clinical and therapeutic data, history of COVID-19 infection, associated symptoms and sequalae. Before moving on to the rest of the questions, participants were questioned about their COVID-19 vaccination status. If the subject had not been vaccinated, the study explored the barriers of not being vaccinated. If the subject had been vaccinated, we inquired about the type of vaccine and the side effects that were associated with the administration of the first and second dose of the COVID-19 vaccine.

Regarding post-vaccination side effects, the participants were asked to choose from symptoms that they experienced within a few days after vaccination. This section was designed to answer questions about general, localized, musculoskeletal, gastrointestinal, psychological, neurological, endocrine, cardiovascular, respiratory, urinary, and allergic symptoms.

The final part concerned the onset of side effects (days from vaccination), timing of maximum symptoms, intensity of symptoms and their effect on activity and need for medical attention. Intensity of symptoms was assessed using the visual analogue scale (VAS), which ranges from 0 (no symptoms at all) to 10 (very severe symptoms).

Patients were approached (in person) and provided complete questionnaire data, as well as examples of commercial names for generic pharmaceutical drugs. The questionnaire was self-administered for people who could read and write, and it was delivered by a researcher for those who could not.

In the final section, those who had not been vaccinated were asked about the barriers to receiving COVID-19 vaccination.

#### Outcome Measures

Our primary outcome was the determination of the vaccination rate among Egyptian HD patients by asking all participants if they had received the COVID-19 vaccine and the secondary outcomes included the identification of adverse events related to COVID-19 vaccination and factors related to these adverse events.

### 2.5. Ethical Consideration

Prior to commencing the study, ethical approval was obtained from the institutional review board of Mansoura University (approval no: R.21.12.1543). The work was carried out in line with the Declaration of Helsinki [12]. The survey was completely voluntary and entirely anonymous. When inviting the participants, the purpose of the study was clearly explained, and informed written consent was obtained.

### 2.6. Statistical Analysis

SPSS (Statistical Product for Social Science) version 20.0 program was used to process the data. Frequencies and relative percentages were used to present the qualitative data. Mean and standard deviation were employed to represent the quantitative data that were normally distributed, while the median, minimum, and maximum were used to represent the quantitative data that were abnormally distributed. Regression analysis was used to determine the factors and associated adverse events related to COVID-19 vaccination in the vaccinated CHD patients. Logistic regression was used because the dependent variable was binary. Two models were considered a univariate model for each covariate and a multivariate model, including gender, age, weight, BMI, smoking, duration of HD, Ca supplementation, associated comorbidities and prior COVID-19 infection.

## 3. Results

### 3.1. Study Participants’ Demographics and General Characteristics

The study included 700 ESRD patients on CHD (response rate of 76.5%), recruited from multiple HD centers in 3 Egyptian governorates. Out of these patients, 30 were excluded due to incomplete data, as shown in the study flow chart (Figure 1).

The mean age of the study participants was 50.79 (SD = 15.9) years. More than half of the patients (58.1%) were females and most of them were married (78.1%). Three hundred and eighty-seven (57.8%) patients were illiterate, 416 (62.1%) were from urban areas, 540 (80.6%) were non-smokers and 417 (62.2%) had an average socioeconomic status. The median duration of HD was 5 years. In terms of comorbidities, 430 (64.2%) patients were hypertensive, 64 (9.6%) were diabetic, and 178 (26.6%) had chronic anemia, as illustrated in Table 1.

### 3.2. Clinical Characteristics of COVID-19 Infection among HD Patients

One hundred and seventy-three patients (25.8%) had previously been infected with COVID-19. Of them, 34 (19.7%) reported prior COVID-19 contact. Fever was the most common symptom of COVID-19 infection (73.4%), followed by cough (35.3%) and fatigue (23.7%). Most of the patients (84.4%) did not need hospitalization nor oxygen supply, while only 6 (3.5%) patients were admitted to an intensive care unit (ICU), as shown in Table 2.

### 3.3. COVID-19 Vaccine Related Anamnesis

After evaluating the advice from healthcare providers (HCP) about receiving the COVID-19 vaccine, 72.2% of patients reported that HCPs strongly endorsed it, 15.1% suggested it, 0.6% of HCPs were unsure, 0.1% advised against it, while only 11.9% reported that HCPs strongly discouraged it.

### 3.4. COVID-19 Vaccination Status

Among the studied 670 CHD patients, only 56 did not receive the COVID-19 vaccine. Fear of the vaccine side effects (28.6%) was the most common cause of not receiving the vaccine, as shown in Figure 2.

On the other hand, the vast majority (614; 91.6%) of the studied patients received two doses of the vaccine. AstraZeneca and Sinopharm were the most used vaccines (50.3% and 35.5%, respectively), as shown in Figure 3.

### 3.5. Adverse Events of COVID-19 Vaccination

The event rate of adverse events associated with COVID-19 vaccination in the vaccinated CHD patients is illustrated in Table 3 and includes the following adverse events.

The main general side effects that were reported after the first dose were generalized weakness/fatigue (56%), headache (43.8%) and fever (40.4%). These manifestations were rarely reported after the second dose. Sore arm/pain was reported in about one third (29.3%) of CHD patients after the first dose, in ten (1.6%) patients after the second dose and in one-hundred-and-three (16.8%) patients after both doses. Localized swelling at the injection site occurred in 32 (5.2%) patients after the first dose. However, localized bleeding was reported in 62 (10.1%) patients after both first and second doses.

Myalgia and generalized muscle pain were reported in 90 (14.7%) patients after the first dose and 9 (1.5%) after both doses. Arthritis/joint pains occurred in only 55 (9%), 2 (0.3%) and 3 (0.5%) patients following the first, second and both doses, respectively. Muscle stiffness/spasm was reported in only one patient following both doses. Decreased appetite was the most common reported gastrointestinal (GI) symptoms (25.1%) in CHD patients after the first dose, followed by abdominal pain (7.5%) and food intolerance. However, GI symptoms rarely occurred after the second dose.

Anxiety (25.4%), psychological stress (12.9%), increase in sleep (9.3%), depression (2.8%) and decrease in memory (2%) were reported among recipients after the first dose and rarely reported after the second one. Tingling at the limb of the injection site (17.1%), sense of incoordination (10.4%), vertigo-like symptoms (8.6%), extremity weakness (3.9%) and reactivation of shingles (0.5%) occurred after the first dose and were seldom reported after the second dose. Of note, no patient reported loss of consciousness/fainting, seizures nor facial weakness following COVID-19 vaccination.

Sore throat (11.2%), nasal stuffiness (8.8%), runny nose (2.8%), ear pain (0.8%), eye pain (0.8%), hoarseness of voice (0.5%) and blurring of vision (0.5%) occurred after the first dose and rarely happened after the second dose. Decreased appetite was reported by 22.5% of patients, increased thirst by 19.2%, heat/cold intolerance by 9.8% and increased appetite by 1.5% after the first dose. However, decreased appetite occurred in 1% of patients, increased thirst in 1%, and heat/cold intolerance in 0.5% after both doses.

Chest pain, blood pressure changes and palpitations were reported in 14.3%, 12.2% and 7.7% of patients, respectively, following the first dose and rarely reported after the second one. Of note, no patient reported syncope after COVID-19 vaccination. Among the respiratory symptoms, shortness of breath occurred in 14.8% of recipients after the first dose, 0.3% after the second one and 0.7% after both doses. Coughing was reported in 1% and 0.5% of patients after the first and second dose. Only one patient reported wheezing after the first dose.

Rash (0.8%), hives (0.7%), swelling in mouth/throat (0.3%), atopic eczema (0.3%) and swelling of lips or tongue (0.2%) were reported after the first dose. In addition, rash and hives were reported in 0.5% and 0.3% of patients, respectively, after both doses.

### 3.6. Other Data about the Adverse Events of COVID-19 Vaccine

As shown in Table 4, adverse events occurred within one day after vaccination. The maximum symptoms usually happened on the second day. The median duration of symptoms was 3 days, with a maximum duration up to 5 days. The median score of VAS used to determine the intensity of symptoms was 3, with a maximum score of 5. About two thirds of vaccinated CHD patients (64.5%) reported some temporary trouble when performing regular daily activities following COVID-19 vaccination. Only 4.2% of patients required transient time off from work. One patient sought help from an emergency department provider and two patients needed hospitalization following COVID-19 vaccination.

### 3.7. Determinants of Adverse Events Related to COVID-19 Vaccination

The univariate logistic regression analysis showed that male gender (OR 1.848; (95% CI, 1.242–2.749), *p* = 0.002), age (OR 0.981; (95% CI, 0.969–0.993), *p* = 0.003), smoking (OR 6.067; (95% CI, 3.514–10.475), *p* < 0.001), duration since starting HD (OR 0.998; (95% CI, 0.998–0.999), *p* < 0.001), calcium supplementation (OR 2.282; (95% CI, 1.517–3.432), *p* < 0.001), antihypertensive medications (OR 3.595; (95% CI, 2.418–5.344), *p* < 0.001), associated comorbidities (OR 2.202; (95% CI, 1.478–3.281), *p* < 0.001) and prior COVID-19 infection (OR 3.318; (95% CI, 1.952–5642), *p* < 0.001) were the main determinants of adverse events related to COVID-19 vaccination, as shown in Table 5.

## 4. Discussion

The COVID-19 pandemic has had a significant negative impact on the dialysis population. SARS-CoV-2 infection and mortality are more common among these patients [13]. These patients are at a higher risk of opportunistic infections, serious sequelae, admission to the ICU, or possibly death due to their old age, uremic status, high comorbidity burden, and poor nutrition [14]. Preventive measures, such as hand hygiene, respiratory hygiene and self-isolation, are applied in HD units to prevent and control the spread of COVID-19 infection [15]. Given the high morbidity and mortality associated with COVID-19, the development of a safe and effective vaccine is critical. The COVID-19 vaccine is a vital milestone in the fight against the virus pandemic [16] and revolutionary in the control of disease among CHD patients [17]. However, safety data on the COVID-19 vaccine in these patients are lacking. This study aimed to determine the vaccination status in Egyptian CHD patients and analyze the safety and detailed side effect profile of the COVID-19 vaccine among these patients. To the best of our knowledge, this is the first study to address this topic in Egyptian CHD patients.

Based on the results of the current study, about one fourth of CHD patients reported previous infection with COVID-19. Most of the patients (84.4%) did not need hospitalization nor oxygen supply, while only 3.5% were admitted to ICU during infection with COVID-19. In a retrospective single center study from China, the incidence of COVID-19 infection among CHD patients was 11% [18]. In a recent meta-analysis of 29 international studies, the incidence of COVID-19 infection in CHD patients was 7.7% and the overall mortality was 22.7% [19]. Variations in viral screening criteria and COVID-19 infection confirmation methods could potentially explain the differences in incidence observed between the studies.

Most of the study patients (91.6%) have received the COVID-19 vaccine. This percentage is a result of the high COVID-19 vaccine acceptance rate in Egypt (90.50%) [20]. Of note, about three fourths of our sample reported that HCPs strongly endorsed it. In dialysis patients, the immune response after vaccination is comparable to that of controls [21]. So, COVID-19 vaccination is critical in this high-risk group in reducing mortality and hospitalization rates [22]. However, some individuals still have unwarranted fears about the vaccine [23], and this explains the percentage of 8.4% of our cohort who refrained from having the vaccination.

Despite the importance of vaccination, we still need more information on the side effects that accompany it, especially in some high-risk groups, including CHD patients. In this study, side effects were more commonly reported after the first dose than the second dose. This is in agreement with the findings of Hatmal and colleagues [24] findings, who showed that the side effects are more obvious after the first dose. On the other side, findings of previous studies [25,26] reported that adverse reactions are more common after administration of the second dose. It would be interesting to include a larger sample of people that received their second dose to compare their experience with side effects to those who received their first dose.

In this study, weakness, fatigue, headache, and fever were the most common reported general side effects of the COVID-19 vaccine. These results are in agreement with those obtained by Kadali and co-authors [27]. In the long term, fever may play a significant role in the immunological response following vaccination; chills and fever were found to have a significant relationship with an antibody response to the SARS-CoV-2 spike protein [28].

The most reported local reaction in our cohort was pain at the injection site. Similarly, in an observational study conducted on 50 vaccinated HD patients, the most prevalent local reaction was pain at the injection site within seven days following the injection, which occurred in 38 % of patients with mild degrees of pain after the first injection [29]. In addition, in a phase 3 trial conducted on 30,420 volunteers to determine the efficacy and safety of the mRNA-1273 SARS-CoV-2 vaccine, 86% reported pain at the injection site. Atypical, very delayed large injection site reactions to RNA COVID-19 vaccines may also occur; they usually begin a week after vaccination, last for several days and are resolved without treatment [30].

Inflammatory musculoskeletal symptoms may occasionally occur in close temporal association with the administration of the COVID-19 vaccine [31]. In a recently published meta-analysis of 15 phase 1/2 clinical trials, myalgia was one of the most common manifestations of the newly licensed vaccines [32]. More than half of the participants reported myalgia after vaccination in a prospective, controlled multicenter study conducted on 159 participants on dialysis, who received 2 doses of the mRNA-1273 COVID-19 vaccine (Moderna) [21]. In the present study, myalgia and generalized muscle pain were found in 14.7% of CHD patients after the first dose and 1.5% after both doses, while arthritis/joint pains occurred in only 9%. There is great uncertainty about whether the COVID-19 vaccine could cause or exacerbate inflammatory arthritis [33,34].

Regarding gastrointestinal (GI) symptoms, about 25% of our cohort reported decreased appetite and 7.5% experienced abdominal pain after the first dose. GI symptoms were rarely reported (0.5%) after COVID-19 vaccine in cancer patients. In general, the majority of GI tract-related side effects observed after COVID-19 vaccination are non-life threatening and transient [35,36].

The most common psychological symptoms reported among our CHD patients after the first dose were anxiety (25.4%), psychological stress (12.9%) and hypersomnolence (9.3%). Impaired sleep quality was reported in 10.65% of health care workers after COVID-19 vaccination [37]. In a survey study from Saudia Arabia conducted on 455 individuals, 3.2% reported a desire to sleep after the second dose compared to only 0.4% after the first dose [38]. In another survey study from Jordon conducted on 2213 participants, sleepiness and laziness were reported in 45.36% of participants [24]. In contrast, Abu-Hammad and coworkers found sleepiness in only 0.2% of a total 409 participants [39].

It is vital to know if COVID-19 vaccinations induce neurologic problems, such as demyelinating illnesses, seizures, or other possible deficits [40]. In real-world data, both vaccinated and unvaccinated people experienced a similar number of neurological events [41]. In the current study, tingling at the limb of the injection site (17.1%), sense of incoordination (10.4%), vertigo-like symptoms (8.6%) and extremity weakness (3.9%) were the most common neurological symptoms associated with the vaccine.

Regarding otolaryngology-related symptoms, sore throat was reported in 11.2% and nasal stuffiness in 8.8% of our patients. This is in agreement with the study by Avci and coauthors [42], in which rhinorrhea (4.4%), sore throat (3.2%), and nasal congestion (2.9%) were the most common otolaryngology-related complaints.

Subacute thyroiditis has been reported in association with COVID-19 vaccination. COVID-19 vaccine adjuvants may provoke an autoimmune/inflammatory condition [43]. Endocrinal symptoms related to COVID-19 vaccination have not been sufficiently highlighted and still need to be clarified.

Kaur and colleagues analyzed the cardiovascular (CV) adverse events reported after the COVID-19 vaccine in the WHO global database. Despite the fact that many CV adverse events were reported after COVID-19 vaccine administration, a causality evaluation is needed to validate the link because these CV adverse events are more common in particular age groups [44].

In the present study, manifestations of allergy, such as a rash, hives and atopic eczema, rarely occurred after the COVID-19 vaccine. Allergy was reported in about one fourth of those who received the COVID-19 vaccine [28]. However, only a few cases of severe allergic reactions have been reported after COVID-19 vaccination [45].

According to our results, the side effects after COVID-9 vaccination in CHD patients are mostly mild to moderate. Only one patient sought help from an emergency department provider and two patients needed hospitalization following COVID-19 vaccination. These findings are in line with earlier work, which found that the majority of the adverse events following COVID-19 vaccination were mild and were related to female sex and young age [46].

Using univariate regression analysis, male gender, age, smoking, duration since starting HD and prior COVID-19 infection were the main determinants of adverse events related to COVID-19 vaccination. In fact, females are more likely than males to experience side effects from vaccination [37]. In a cross-sectional survey study conducted in the United Arab Emirates to collect data on the effects of the COVID-19 vaccine on individuals, after the first dose of vaccination, 83% of females experienced adverse effects compared to 55% of men, and 98.5% of females versus 80% of males after the second dose [47]. The difference in side effects recorded for inactivated virus vaccinations, such as influenza, the measles, mumps and rubella combination vaccine, and attenuated dengue vaccines, across genders suggests that females have higher immune responses and side effects are more frequent [48]. Short-term experiments with Pfizer-BioNTech mRNA vaccines found that older adults had poorer immune responses [49]. In addition to lower post-vaccine responses, older people showed faster waning of antibodies following vaccinations [50]. However, we report in our study that older individuals were associated with more adverse events related to the COVID-19 vaccine.

To the best of our knowledge, this is the first study to investigate the side effects of the COVID-19 vaccine in CHD patients in Egypt. The strength of this study is the relatively large number of participants. However, certain limitations should be acknowledged. First, we employed self-reported data, which might lead to information bias, such as inconsistency in participants’ reports. Second, this study only investigated the vaccine’s short-term side effects (immediately after injection). The vaccine’s medium and long-term effects are still unknown. Third, patients received different types of COVID-19 vaccine, which precludes conclusion about the side effect of a specific type. Multi-nation, larger sample size research, including a more diversified population, as well as assessing probable long-term side effects of various COVID-19 vaccinations, is recommended.

## 5. Conclusions

In conclusion, the side effects after COVID-9 vaccination in CHD patients are mostly mild to moderate. Male gender, age, smoking, duration since starting HD and prior COVID-19 infection were the main determinants of adverse events related to COVID-19 vaccination. Our preliminary findings support the favorable short-term safety profile of the COVID-19 vaccine among CHD patients, and hence can reassure both clinicians and patients, as well as further promote COVID-19 vaccine administration among these patients.

## Figures and Tables

**Figure 1 vaccines-10-01771-f001:**
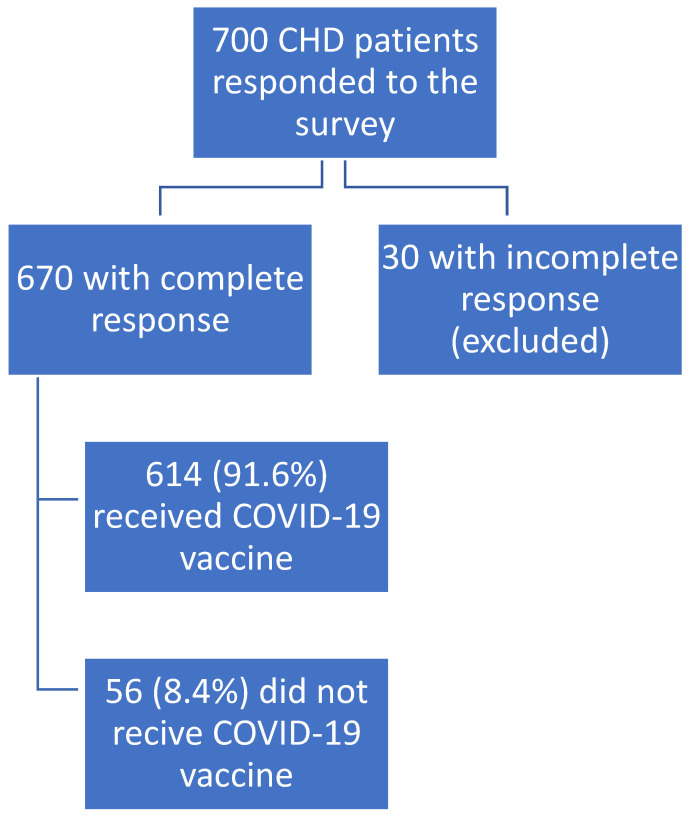
Classification of the survey responses from patients on chronic hemodialysis (CHD).

**Figure 2 vaccines-10-01771-f002:**
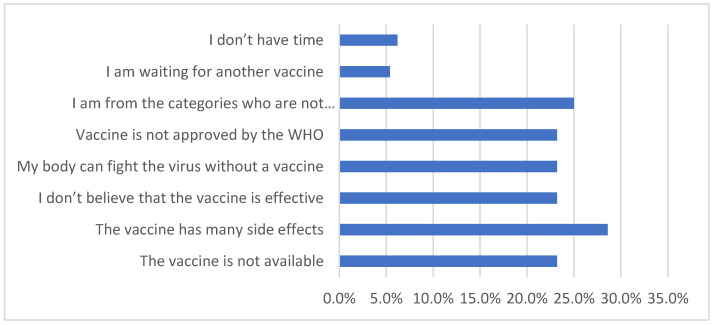
Barriers to receive COVID-19 vaccine in vaccine non-recipients (n = 56).

**Figure 3 vaccines-10-01771-f003:**
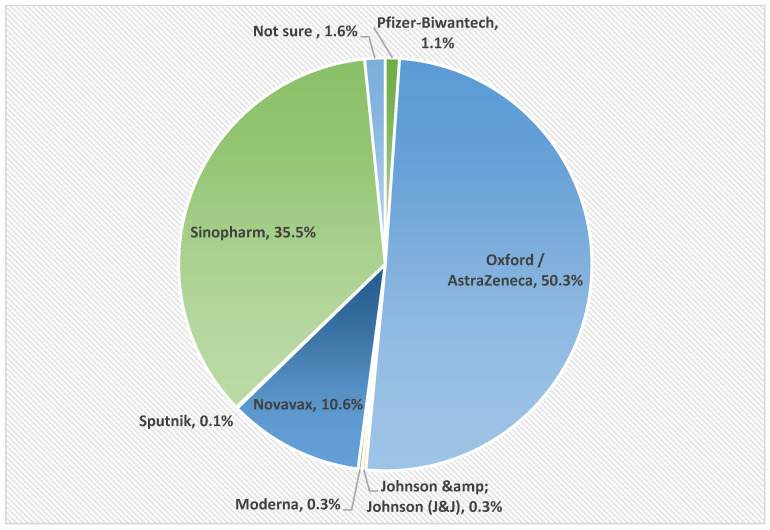
Type of vaccine COVID-19 administrated in CHD patients who received the vaccine (n = 614).

**Table 1 vaccines-10-01771-t001:** Sociodemographic data and clinical characteristics of the studied CHD patients (n = 670).

Variable Mean ± SD, n (%)	HD Patients (n = 670)
Demographic data	
Gender	
Male	389 (58.1)
Female	281 (41.9)
Age (years)	50.79 ± 15.9
Marital status	
Single	77 (11.5)
Married	523 (78.1)
Widowed	58 (8.7)
Divorced	12 (1.8)
Education level	
Illiterate	387 (57.8)
Primary	85 (12.7)
High school/diploma	145 (21.6)
College level	47 (7)
Postgraduate (masters/doctorate)	6 (0.9)
Employment status	
Unemployed	513 (76.6)
Employed	83 (12.4)
Retired	68 (10.1)
Student	6 (0.9)
Residence	
Urban	416 (62.1)
Rural	254 (37.9)
Smoking habit	
Non-smoker	540 (80.6)
Smoker	85 (12.7)
Ex-smoker	45 (6.7)
Socioeconomic status	
Low	239 (35.7)
Average	417 (62.2)
High	14 (2.1)
Anthropometric measures	
Weight (Kg)	73.94 ± 20.21
Height (m)	163.07 ± 11.25
BMI (Kg/m^2^)	27.76 ± 7.17
Clinical data	
Duration of hemodialysis (years)	5 (2–10)
Associated comorbidities	
No	166 (24.8)
Diabetes	64 (9.6)
Hypertension	430 (64.2)
Chronic anemia	178 (26.6)
Heart disease	26 (3.9)
Liver disease	3 (0.4)
Chronic respiratory disease	4 (0.6)
Autoimmune disease	2 (0.3)
Cancer	14 (2.1)
Therapeutic data	
None	66 (9.9)
Erythropoietin	522 (77.9)
Iron supplementation	455 (67.9)
Calcium supplementation	479 (71.5)
Alpha calcidol	483 (72.1)
Calcimimetics	26 (3.9)
Aluminum hydroxide	61 (9.1)
Antihypertensive drugs	374 (55.8)
Antidiabetic drugs	37 (5.5)
Immunosuppressive drugs	1 (0.1)

**Table 2 vaccines-10-01771-t002:** Clinical characteristics of COVID-19 infection among the studied HD patients (n = 173).

Variable	HD Patients with Prior COVID-19 Infection (n = 173) n (%)
Prior COVID-19 contact	34 (19.7)
*Symptoms associated with COVID-19 infection*	
Fever	127 (73.4)
Fatigue	41 (23.7)
Cough	61 (35.3)
Skin rash	2 (1.2)
Diarrhea	20 (11.6)
Pneumonia	18 (10.4)
Dyspnea	29 (16.8)
Headache	30 (17.3)
Chest pain	14 (8.1)
Oral ulcers	5 (2.9)
Anorexia/vomiting	17 (9.8)
Joint pain	18 (10.4)
*Hospitalization during COVID-19 infection*	
No, with no need for oxygen	146 (84.4)
No, with a need for oxygen	3 (1.7)
Yes, with no need for oxygen	19 (11)
Yes, with a need for oxygen	5 (2.9)
ICU admission during COVID-19 infection	6 (3.5)
Duration of symptoms related to COVID-19 infection (days)	3 (2–7)

**Table 3 vaccines-10-01771-t003:** Event rate of adverse events associated with COVID-19 vaccination in the vaccinated CHD patients (n = 614).

Variable	Vaccinated CHD Patients (n = 614)		
	After the First Dose Only n (%)	After the Second Dose Only n (%)	After the First and Second Doses n (%)
General side effects			
Generalized weakness/fatigue	344 (56)	4 (0.7)	20 (3.3)
Headache	269 (43.8)	4 (0.7)	8 (1.3)
Fever	248 (40.4)	4 (0.7)	4 (0.7)
Localized symptom/s			
Sore arm/pain	180 (29.3)	10 (1.6)	103 (16.8)
Localized swelling at the injection site	32 (5.2)	9 (1.5)	2 (0.3)
Itching	22 (3.6)	1 (0.2)	4 (0.7)
Lymphadenopathy (axillary or regional)	3 (0.5)	0	0
Rash	3 (0.5)	0	1 (0.2)
Musculoskeletal symptom/s			
Muscle pain/myalgia	90 (14.7)	2 (0.3)	9 (1.5)
Arthritis/joint pains	55 (9)	2 (0.3)	3 (0.5)
Muscle stiffness/spasm	0	0	1 (0.2)
Gastrointestinal symptom/s			
Nausea	25 (4.1)	1 (0.2)	2 (0.3)
Decreased appetite	154 (25.1)	1 (0.2)	4 (0.7)
Diarrhea	13 (2.1)	2 (0.3)	1 (0.2)
Abdominal pain	46 (7.5)	4 (0.7)	1 (0.2)
Vomiting	3 (0.5)	0	0
Heartburn	25 (4.1)	4 (0.7)	3 (0.5)
Psychological and/or psychiatric symptom/s			
Feelings of joy/relief/gratitude	7 (1.1)	0	3 (0.5)
Anxiety	156 (25.4)	1 (0.2)	3 (0.5)
Increase in sleep	57 (9.3)	0	6 (1)
Psychological stress	79 (12.9)	0	1 (0.2)
Decrease in memory	12 (2)	0	1 (0.2)
Neurological symptom/s			
Brain fogging or reduced mental clarity	10 (1.6)	0	3 (0.5)
Tingling	105 (17.1)	0	4 (0.7)
Vertigo-like symptoms	53 (8.6)	1 (0.2)	0
Paralysis/extremity weakness	24 (3.9)	0	1 (0.2)
Incoordination	64 (10.4)	0	3 (0.5)
Reactivation of shingles	3 (0.5)	0	0
Head/ear/eyes/nose/throat symptom/s			
Nasal stuffiness	54 (8.8)	1 (0.2)	0
Sore throat	69 (11.2)	0	1 (0.2)
Runny nose	17 (2.8)	2 (0.3)	0
Ringing sensation in ears	4 (0.7)	1 (0.2)	1 (0.2)
Ear pain	5 (0.8)	0	0
Eye pain	5 (0.8)	1 (0.2)	1 (0.2)
Blurring of vision	3 (0.5)	0	1 (0.2)
Endocrine symptom/s			
Decreased appetite	138 (22.5)	1 (0.2)	6 (1)
Heat/cold intolerance	60 (9.8)	1 (0.2)	3 (0.5)
Increased thirst	118 (19.2)	0	6 (1)
Increased appetite	9 (1.5)	1 (0.2)	0
Cardiovascular symptom/s			
Palpitations	47 (7.7)	1 (0.2)	2 (0.3)
Chest pain	88 (14.3)	0	5 (0.8)
Blood pressure changes	75 (12.2)	1 (0.2)	10 (1.6)
Respiratory symptom/s			
Shortness of breath	91 (14.8)	2 (0.3)	4 (0.7)
Cough	6 (1)	3 (0.5)	0
Wheezing	1 (0.2)	0	0
Allergic symptom/s			
Rash	5 (0.8)	0	3 (0.5)
Hives	4 (0.7)	0	2 (0.3)
Swelling in mouth/throat	2 (0.3)	0	0
Atopic eczema	2 (0.3)	0	0
Swelling of lips or tongue	1 (0.2)	0	0

**Table 4 vaccines-10-01771-t004:** Data about the adverse events of COVID-19 vaccine in the vaccinated CHD patients (n = 614).

Variable n (%), Median (min–max)	Vaccinated CHD Patients (n = 614)
*Intensity of symptoms (VAS)*	3 (0–5)
*Timing of adverse events*	
Onset of symptoms post-vaccine (days)	1 (1–2)
Timing of maximum symptoms (days after vaccination)	2 (1–3)
Duration of symptoms (days) (patency period)	3 (1–5)
*Activity or need for medical attention*	
Trouble to perform regular daily living activities temporarily	396 (64.5)
Required transient time off from work	26 (4.2)
Required to seek help from outpatient provider	3 (0.5)
Required to seek help from emergency department provider	1 (0.2)
Required hospitalization and subsequent inpatient care	2 (0.3)

**Table 5 vaccines-10-01771-t005:** Univariate and multivariate regression analysis used to determine factors associated with adverse events related to COVID-19 vaccination in the vaccinated CHD patients (n = 614).

Variable	Univariate			Multivariate		
	OR	95% CI	*p*	OR	95% CI	*p*
Constant	-			70.586		0.000
Male gender	1.848	1.242–2.749	0.002 *	0.793	0.284–2.215	0.658
Age	0.981	0.969–0.993	0.003 *	0.959	0.936–0.983	0.001 *
Weight	0.996	0.987–1.005	0.412	0.962	0.921–1.004	0.079
BMI	1.005	0.976–1.034	0.759	1.156	1.029–1.298	0.014 *
Residence (urban)	6.629	4.408–9.967	<0.001 *	0.809	0.380–1.724	0.583
Smoking	6.067	3.514–10.475	<0.001 *	0.203	0.081–0.512	0.001 *
Duration since starting HD	0.998	0.998–0.999	<0.001 *	0.999	0.998–0.999	<0.001 *
Calcium supplementation	2.282	1.517–3.432	<0.001 *	0.415	0.178–0.967	0.042 *
Antihypertensive medications	3.595	2.418–5.344	<0.001 *	0.403	0.196–0.826	0.013 *
Associated comorbidities	2.202	1.478–3.281	<0.001 *	0.330	0.139–0.783	0.012 *
Prior COVID-19 infection	3.318	1.952–5.642	<0.001 *	1.295	0.531–3.158	0.570

* *p* < 0.05.

## Data Availability

All data generated or analyzed during this study are included in this published article.

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
