# Peer review of "COVID-19 Vaccination Trends and Side Effects among Egyptian Hemodialysis Patients: A Multicenter Survey Study"

_vaccines, 2022, doi:10.3390/vaccines10101771_

Round 1
Reviewer 1 Report
The manuscript by Nassar et al. is a survey-based study about the vaccination of COVID-19 on the CHD. This article is not of great clinical importance, the current content is not well written, however I suggest to use more references in their paper and use similiat previous article about questionnaire and COVID-19. This MS might have some scientifically error. Nonetheless, there are a couple of limitations in results section that unfortunately prevented me from recommending acceptance. in results, 670 participants included in this study but only 614 participants vaccinated and the questionnaire is about the participants who received vaccine, therefore analysis must be based on 614 participants not 670.
The materials & methods section is relatively immature. You could expand it a bit more clearly for readers. For example, write about the sample size. How did you calculate sample size for this survey? Furthermore, discuss more about your sampling strategy.
Write about the validity and reliability of the measurement tools (CVR and CVI).
Mention the possible score (range) for each scale and meaning of it so easier to readers interpret the results.
It seems that some variables have compared to other variables and need to use statistical tests such as chi-square and spearman's rank test. In the case of non-normal data, Median and IQR better present location and dispersion of data.
In addition, logistic regression analysis and comparison of means are needed.
Minor comments:
line 77: please clearly mention the name of centers, city and country
in figure 1: 56 (8.4%) did not receive COVID-19 vaccine
Author Response
Reply to reviewer 's comments
Dear respected editor and reviewer,
We wish to thank you all for your constructive comments on this manuscript. Your comments provided valuable insights to refine its contents and analysis. In this document, we try to address the issues raised as best as possible
Ms. Ref. No.: Vaccines-1952219
Title: COVID-19 vaccination trends and side effects among Egyptian hemodialysis patients: a multicenter survey study
The manuscript by Nassar et al. is a survey-based study about the vaccination of COVID-19 on the CHD. This article is not of great clinical importance, the current content is not well written, however I suggest using more references in their paper and use similar previous article about questionnaire and COVID-19.
Reply The comments were noted, and the manuscript revised and rearranged at all sections and more references were added
This MS might have some scientifically error. Nonetheless, there are a couple of limitations in results section that unfortunately prevented me from recommending acceptance. in results, 670 participants included in this study but only 614 participants vaccinated, and the questionnaire is about the participants who received vaccine; therefore, analysis must be based on 614 participants not 670.
Reply No scientific error included. We included all patients on HD either vaccinated or not and then we studied the side effects occurred to those vaccinated .On the other hand ,we evaluated the barriers for vaccination on those who did not receive vaccine .We already added a flowchart to illustrate this point as the following.
The materials & methods section is relatively immature. You could expand it a bit more clearly for readers. For example, write about the sample size. How did you calculate sample size for this survey? Furthermore, discuss more about your sampling strategy.
Reply The materials and methods section was edited and modified
“The study included ESRD patients on CHD for more than 6 months and aged >18 years either were vaccinated with COVID-19 vaccine or not Vaccinated participants must have gotten the most recent dosage of the vaccine, either the first or second dose, no more than thirty days before completing the questionnaire.”
Patients were approached (in person) and provided complete questionnaire data as well as examples of commercial names for generic pharmaceutical drugs. The questionnaires were given to patients at respective HD units by the researchers.The questionnaire was self-administered for people who could read and write, and it was delivered by a researcher for those who couldn't.
Calculation of sample Size
The online sample size calculator RaoSoft® was used to determine the appropriate sample size. Based on an estimated prevalence of patients on dialysis is 264 pmp [1], 50% predicted response, 5% margin of error and 95% confidence level, the minimum sample size was 397 participants.”
Write about the validity and reliability of the measurement tools (CVR and CVI).
Reply
The reliability of the questionnaire was assessed as the following and this point was discussed as the following
“Five nephrology staff members examined the translation, questionnaire design, substance, words, comprehension, and ease of completion in a pilot study to validate the questionnaire. The questionnaire was finalized through an iterative conversation. The questions were written to be as straightforward as feasible. Answers were limited to yes/no questions (closed-ended questions). Later, the questionnaire's reliability was assessed by a group of twenty recently vaccinated HD patients, who completed the questionnaire twice with a minimum interval of two weeks. The provisional instrument's re-test resulted in significant reliability, with a mean Cohen's kappa coefficient of 0.84 ±0.16.”
Mention the possible score (range) for each scale and meaning of it so easier to readers interpret the results.
Reply The use scale is visual analogue scale, and its range was determined “visual analogue scale (VAS) which scored from 0 (no symptoms at all) to 10 (very severe symptoms)’
It seems that some variables have compared to other variables and need to use statistical tests such as chi-square and spearman's rank test. In the case of non-normal data, Median and IQR better present location and dispersion of data.
Reply we did not compare groups .We only added description of data in the form of “Frequencies and relative percentages were used to present qualitative data. Mean and standard deviation were employed to represent quantitative data that was normally distributed while median, minimum, and maximum were used to represent quantitative data that was abnormally distributed. Regression analysis was used to determine factors associated adverse events related to COVID-19 vaccination in the vaccinated CHD patients. Logistic regression was used because the dependent variable was binary.Two model was considered ; the first one ,univariate for each covariate and the second one ,multivariate including gender ,age ,weight ,BMI ,smoking ,duration of HD ,Ca supplementation ,associated comorbidities and prior COVID-19 infection .”
In addition, logistic regression analysis and comparison of means are needed.
Reply Logistic regression analysis was added and described in results and discussion
Table 5. Univariate and multivariate regression analysis to determine factors associated with adverse events related to COVID-19 vaccination in the vaccinated CHD patients (n= 614)
Variable |
Univariate |
Multivariate |
||||
OR |
95% CI |
P |
OR |
95% CI |
P |
|
Constant |
- |
70.586 |
|
0.000 |
||
Male gender |
1.848 |
1.242-2.749 |
0.002* |
0.793 |
0.284-2.215 |
0.658 |
Age |
0.981 |
0.969-0.993 |
0.003* |
0.959 |
0.936-0.983 |
0.001* |
Weight |
0.996 |
0.987-1.005 |
0.412 |
0.962 |
0.921-1.004 |
0.079 |
BMI |
1.005 |
0.976-1.034 |
0.759 |
1.156 |
1.029-1.298 |
0.014* |
Residence (Urban) |
6.629 |
4.408-9.967 |
<0.001* |
0.809 |
0.380-1.724 |
0.583 |
Smoking |
6.067 |
3.514-10.475 |
<0.001* |
0.203 |
0.081-0.512 |
0.001* |
Duration since starting HD |
0.998 |
0.998-0.999 |
<0.001* |
0.999 |
0.998-0.999 |
<0.001* |
Calcium supplementation |
2.282 |
1.517-3.432 |
<0.001* |
0.415 |
0.178-0.967 |
0.042* |
Antihypertensive medications |
3.595 |
2.418-5.344 |
<0.001* |
0.403 |
0.196-0.826 |
0.013* |
Associated comorbidities |
2.202 |
1.478-3.281 |
<0.001* |
0.330 |
0.139-0.783 |
0.012* |
Prior COVID-19 infection |
3.318 |
1.952-5.642 |
<0.001* |
1.295 |
0.531-3.158 |
0.570 |
*p<0.05
Minor comments:
line 77: please clearly mention the name of centers, city and country
Reply We added these required data .
in figure 1: 56 (8.4%) did not receive COVID-19 vaccine
Reply We illustrated that the aim of the staudy is to assess who the vaccination status .So we included all HD patients and the divided them to the vaccinated and non-vaccinated participants and we illustrated that at the materials and methods and the flow chart of the study
If the reviewer is not satisfied, we can consider including more literature and give more details of our study. If the language is still not clear enough, we would be grateful if in the next round of revisions, you point to us the specific sentences we should improve
Once again, we thank you for the time you put in reviewing our paper and look forward to meeting your expectations. Since your inputs have been precious, in the eventuality of a publication, we would like to acknowledge your contribution explicitly.

Reviewer 2 Report
This is an interesting manuscript on COVID-19 vaccination research.
Please consider the following changes:
1. The abstract is very extensive. Please consider some changes to focus only on the most important issues.
2. Please add more information on COVID-19 in Egypt, access to COVID-19 vaccines, and other related issues that would be important for international readers.
3. Methods - please provide more informative data on patient enrolment. How many facilities, inclusion/exclusion criteria, etc.
4. Please provide data on the response rate and the representativeness of the population.
5. The results section requires extensive revisions. There is no need to divide results into over 12 sub-section. Please focus on the most important findings and present other data in figures/tables.
6. The scope of statistical analysis is very simple. Please provide more advanced methods.
7. There is a lack of clearly defined "take-home message". Please revise the manuscript to provide a clearly defined "take-home message" and implications of this study.
8. 2-3 sentences on the practical implications of this study will be interesting.
Author Response
Reply to reviewer 's comments
Reply to reviewer 's comments
Dear respected editor and reviewer,
We wish to thank you all for your constructive comments on this manuscript. Your comments provided valuable insights to refine its contents and analysis. In this document, we try to address the issues raised as best as possible
Ms. Ref. No.: Vaccines-1952219
Title: COVID-19 vaccination trends and side effects among Egyptian hemodialysis patients: a multicenter survey study
This is an interesting manuscript on COVID-19 vaccination research.
Reply Thank you for your valuable comments and encouraging words
Please consider the following changes:
The abstract is very extensive. Please consider some changes to focus only on the most important issues.
Reply Noted and changes were done to focus only on the most important issues as the following
“Abstract
Background: Vaccination may be a key intervention to prevent infection in chronic hemodialysis (CHD) patients. This study aimed to determine the COVID-19 vaccination status in Egyptian CHD patients and to analyze the safety and detailed side effect profile of COVID-19 vaccine among these patients.
Methods:
This survey-based study conducted on 670 end stage renal disease (ESRD) patients on CHD at the period from 3 December 2021 to 5 February 2022. Subjects were asked about sociodemographic characteristics, clinical and therapeutic data in addition to their COVID-19 vaccination status. If the subject had been vaccinated, we inquired about the type of vaccine and the side effects that occurred within few days after administrating of the first and second dose of COVID-19 vaccine. Additionally, subjects were inquired about the onset of side effects (days from vaccination), timing of maximum symptoms, intensity of symptoms and their effect on activity and need for medical attention.
Results:
The study included 670 CHD patients with mean age of 50.79 years ,58.1% were females. The vast majority (614; 91.6%) of the studied patients received 2 doses of the vaccine. Side effects were commonly reported after the first dose than the second dose. The main side effects that reported were generalized weakness/fatigue (56%), headache (43.8%) and fever (40.4%), , and sore arm/pain was reported (29.3%). Adverse events mostly occurred within one day after vaccination and the maximum symptoms usually happened at the second day. The median duration of symptoms was 3 days with maximum duration up to 5 days. The univariate logistic regression analysis showed that male gender (OR 1.848; [95% CI, 1.242-2.749], p =0.002), age (OR 0.981; [95% CI, 0.969-0.993], p =0.003), smoking (OR 6.067; [95% CI, 3.514-10.475], p <0.001), duration since starting HD (OR 0.998; [95% CI, 0.998-0.999], p <0.001), associated comorbidities (OR 2.202; [95% CI, 1.478-3.281], p <0.001), prior COVID-19 infection (OR 3.318; [95% CI, 1.952-5.642], p <0.001) were the main determinates of adverse events related to COVID-19 vaccination
Conclusions:
Our preliminary findings support the favorable short term safety profile of COVID-19 vaccine among CHD patients and hence, can reassure both clinicians and patients, as well as promote further COVID-19 vaccine administration among these patients. “
- Please add more information on COVID-19 in Egypt, access to COVID-19 vaccines, and other related issues that would be important for international readers.
Reply
We agree with the reviewer regarding this point. So ,we added a paragraph on this point at the end of the introduction as the following “The Egyptian government has devised and executed plans to contain and mitigate this pandemic's effects on human health and the economy [8]. It initiated a number of programmes to increase public awareness of the worldwide epidemiological crisis [9]. Parallel to these initiatives, all citizens are being offered with free vaccinations with priority given to healthcare workers (HCWs) and elderly individuals, particularly those with chronic conditions [10]. In this regard, the government has also made arrangements to get vaccinations from various sources to meet its immunization requirements [11].”
- Methods - please provide more informative data on patient enrolment. How many facilities, inclusion/exclusion criteria, etc.
Reply we added more details including patient enrolment and inclusion and exclusion criteria as the following “The study included ESRD patients on CHD for more than 6 months and aged >18 years either were vaccinated with COVID-19 vaccine or not Vaccinated participants must have gotten the most recent dosage of the vaccine, either the first or second dose, no more than thirty days before completing the questionnaire”
“Patients were approached (in person) and provided complete questionnaire data as well as examples of commercial names for generic pharmaceutical drugs. The questionnaires were given to patients at respective HD units by the researchers. The questionnaire was self-administered for people who could read and write, and it was delivered by a researcher for those who couldn't. “
Calculation of sample Size
The online sample size calculator RaoSoft® was used to determine the appropriate sample size. Based on an estimated prevalence of patients on dialysis is 264 pmp [1], 50% predicted response, 5% margin of error and 95% confidence level, the minimum sample size was 397 participants.
“The questionnaire was finalized through an iterative conversation. The questions were written to be as straightforward as feasible. Answers were limited to yes/no questions (closed-ended questions). Later, the questionnaire's reliability was assessed by a group of twenty recently vaccinated HD patients, who completed the questionnaire twice with a minimum interval of two weeks. The provisional instrument's re-test resulted in significant reliability, with a mean Cohen's kappa coefficient of 0.84 ±0.16”.
- Please provide data on the response rate and the representativeness of the population.
Reply We added the method of sample size calculation as the following “Calculation of sample Size
The online sample size calculator RaoSoft® was used to determine the appropriate sample size. Based on an estimated prevalence of patients on dialysis is 264 pmp [1], 50% predicted response, 5% margin of error and 95% confidence level, the minimum sample size was 397 participants.”
Also, we added the response rate as the following
“The study included 700 ESRD patients on CHD (response rate ,76.5%) “
- The results section requires extensive revisions. There is no need to divide results into over 12 sub-section. Please focus on the most important findings and present other data in figures/tables.
Reply
We have revised the tables and delete a lot of minor results. Also ,we summarized the results section as possible.
- The scope of statistical analysis is very simple. Please provide more advanced methods.
Reply We added sample size calculation and univariate and multivariate regression analysis as the following “Regression analysis was used to determine factors associated adverse events related to COVID-19 vaccination in the vaccinated CHD patients. Logistic regression was used because the dependent variable was binary. Two model was considered; the first one ,univariate for each covariate and the second one ,multivariate including gender ,age ,weight ,BMI ,smoking ,duration of HD ,Ca supplementation ,associated comorbidities and prior COVID-19 infection ‘
- There is a lack of clearly defined "take-home message". Please revise the manuscript to provide a clearly defined "take-home message" and implications of this study. 8. 2-3 sentences on the practical implications of this study will be interesting.
Reply
We added this sentences in the conclusion
“The side effects after COVID-9 vaccination in CHD patients are mostly mild to moderate. male gender, age, smoking, duration since starting HD and prior COVID-19 infection were the main determinants of adverse events related to COVID-19 vaccination. Our preliminary findings support the favorable short term safety profile of COVID-19 vaccine among CHD patients and hence, can reassure both clinicians and patients, as well as promote further COVID-19 vaccine administration among these patients. “
If the reviewer is not satisfied, we can consider including more literature and give more details of our study. If the language is still not clear enough, we would be grateful if in the next round of revisions, you point to us the specific sentences we should improve
Once again, we thank you for the time you put in reviewing our paper and look forward to meeting your expectations. Since your inputs have been precious, in the eventuality of a publication, we would like to acknowledge your contribution explicitly.

Round 2
Reviewer 1 Report
line 106: an iterative Theconversation?
Reviewer 2 Report
The mansucript was significantly revised and can be considered for publication in present form.